# Efficacy Evaluation of Chlorine Dioxide and Hypochlorous Acid as Sanitisers on Quality and Shelf Life of Atlantic Salmon (*Salmo salar*) Fillets

**DOI:** 10.3390/foods13193156

**Published:** 2024-10-03

**Authors:** Wing H. Chung, Md Reaz Chaklader, Janet Howieson

**Affiliations:** 1School of Molecular and Life Sciences, Faculty of Science and Engineering, Curtin University, Bentley, WA 6102, Australia; reaz.chaklader@dpird.wa.gov.au (M.R.C.); j.howieson@curtin.edu.au (J.H.); 2Department of Primary Industries and Regional Development, Fleet Street, Fremantle, WA 6160, Australia

**Keywords:** quality index, dipping, chlorine ice, infused ice, physicochemical quality, simulated retail display, seafood

## Abstract

Microbial contamination during seafood processing can often lead to a reduction in shelf life and the possibility of food-borne illnesses. Sanitisation with chlorine-based products during seafood processing is therefore sometimes undertaken. This study compared the effects of two sanitisers, chlorine dioxide (ClO_2_) and hypochlorous acid (HOCl) at their suggested concentration (5 ppm and 10 ppm; 50 ppm and 100 ppm respectively), on physical, chemical, and microbial qualities of Atlantic salmon (*Salmo salar*) fillets throughout 7 days of simulated retail display refrigeration. Parameters used for assessment included quality index (QI), drip loss, colour, texture, histology, total volatile base nitrogen (TVB-N), lipid oxidation (malonaldehyde, MDA), pH, and total viable count changes. Results indicated that whilst drip loss increased over the storage time, day 4 and 7 drip loss in both sanitisers decreased significantly compared with the control. There was a linear relationship (R > 0.70) between QI and storage time in all treatments, particularly in regard to skin brightness, flesh odour, and gaping parameters, but treatment differences were not present. Texture parameters including gumminess, chewiness, and hardness increased over time in the control whilst both sanitiser treatments seemed to provide protective effects against texture hardening during storage. The observed softening effects from the sanitiser treatments were aligned with microstructural and cytological changes in the histology results, as evidenced by a reduced fibre–fibre adhesion, myodigeneration, and an increase in interfibrillar space over storage time. Colour, especially chroma (C*), was shown to decrease over time in control, whereas insignificant protective effects were observed in both sanitiser treatments at day 7. Irrespective of treatment and storage time, MDA levels exceeded the acceptable limit on all days, whilst TVB-N levels were below the critical limit. Although pH was influenced by treatment and storage time, the pH was within the normal range. Microbiological results showed that with sanitiser addition, TVC was below the permissible level (10^6^ CFU/g) until day 4 but ClO_2_ ice (5 ppm), ClO_2_ (10 ppm), and HOCl (100 ppm) treated fillets all exceeded the limit on day 7. The mixed results on the effect of sanitiser addition on fillet quality and shelf life suggested that further investigation on pathogen reduction, sanitiser introductory method, as well as testing the same treatments in low-fat fish models would be recommended.

## 1. Introduction

Microbial contamination can happen during pre-, post- and harvest processing in seafood, particularly in fish because of the presence of high levels of protein, polyunsaturated fatty acids, free amino acids, and micronutrients. These nutrients support the growth of microorganisms originating from the resident microbial flora, equipment surfaces, workers during processing, and aerial sources. Contamination during food processing could lead to potential food-borne illnesses and result in microbiological spoilage, causing off-odours, off-flavours, slime, and discolouration. An outcome is the reduction in shelf life of seafood products [1,2]. Approximately one in ten people around the world fall ill every year after eating contaminated food, and seafood has been identified to be one of the high-risk foods that contribute to this [3,4]. Atlantic salmon (*Salmo salar*) is one of the popular seafood products worldwide that are often sold fresh and are commonly eaten raw among certain cultures, making it an excellent vehicle for microorganisms [5,6,7]. The beneficial polyunsaturated fatty acids, including omega-3 long-chain fatty acids such as eicosapentaenoic acid (EPA) and docosahexaenoic acid (DHA), are susceptible to oxidation even at refrigeration temperatures, which has made salmon a highly perishable product with a very short storage life [8]. Hence, techniques to maintain its organoleptic characteristics have been a main focus of salmon industries to bring fresh fish to the consumer [9].

To maintain safety, quality and reduce spoilage of seafood products, post-harvesting technology has been implemented to enhance microbial quality in fresh fish [1]. Sanitisation with sodium hypochlorite (NaOCl) is a widely used processing tools in the fish industry. However, even though it is effective in removing microorganisms, there is a major downside to using this strong chlorine-based oxidant. For instance, it reacts with organic materials to form organochlorines such as trihalomethanes (THM), which is a potential carcinogen that could lead to adverse health outcomes [1,2]. As a consequence, there is a need for research into and development of effective alternative sanitisers that have less impact on both the fish products and consumers [1].

Among the replacements, chlorine-based sanitisers such as calcium hypochlorite (Ca(ClO)_2_) (granular or powdered form) are commonly used in seafood processing before packaging and distribution [10,11] due to their low-cost and broad-spectrum bactericidal activity. However, the use of Ca(ClO)_2_ on the edible portion of fish and shellfish is limited because of diverse opinions and regulations across different countries. Chlorine dioxide (ClO_2_) and hypochlorous acid (HOCl) are some of the chlorine-based oxidants that have been shown to be less harmful than NaOCl yet demonstrate a similar level of effectiveness. In a previous study by Lin et al. [12], CIO_2_ was shown to be more potent than aqueous chlorine in killing bacteria and less reactive with organic material. A relatively smaller dosage of the chemical could be used to achieve the same level of sanitisation while limiting alteration to the food products. Alternatively, HOCl is more active in killing microorganisms than NaOCl; a lesser quantity is needed to achieve ideal microbial quality. HOCl is approved by USDA-FSIS [13] to be a safe and suitable ingredient in a limited dosage and more economical than other chlorine-based products [2,14]. The high oxidation-reduction potential has been reported to kill microorganisms by penetrating the outer bacterial membrane, destroying proteins and enzymes involved in metabolic processes. However, the effectiveness of chlorine-based sanitisers depends on the pH [11]. Nearly neutral pH has been reported to be more reactive, whilst a low pH has caused a loss in activity through off-gassing and degradation of free chlorine [11].

There is a plethora of studies on the application of ClO_2_ and HOCl in plant-based products [15,16,17,18,19], though the majority of those conducted so far on fish or seafood were mostly at one time point without considering the changes in physicochemical and microbial quality throughout an extended period [12,14,20,21,22,23,24,25,26,27]. The remaining studies that have investigated the effects of these sanitisers throughout their shelf life did not compare the effects of more than one chlorine-based sanitiser [24,28,29,30,31]. For example, 45 ppm and 150 ppm of HOCl-containing water-based sanitisation products decreased the total bacterial load and specific spoilage organisms and extended the shelf life of Southern Australian King George Whiting and Tasmanian Atlantic Salmon fillets [32]. It has been difficult to compare the effects of alternative chlorine-based sanitisers on fish before reaching consumers.

Thus, this study aims to examine and compare the effects of ClO_2_ and HOCl on fish by monitoring physical, chemical, and microbiological quality changes using Atlantic salmon as a model throughout 7 days of simulated retail display refrigeration.

## 2. Materials and Methods

### 2.1. Raw Materials

Eight Atlantic salmon (*Salmo salar*) from Dover (TSA, Australia) were purchased via a local seafood wholesaler (Belmont, WA, Australia) and filleted professionally. Skin-on fillets obtained between the initium of the dorsal fin and anal fin from salmons were then divided into 72 portions ranging from 100 to 150 g, including the dorsal and ventral muscles. Samples were chilled on ice inside a polystyrene box immediately post-portioning and transported within one hour to Curtin Aquatic Research Laboratories (CARL) (Bentley, WA, Australia). On arrival, all the portions were mixed into a 3.5 % ice-cold sterile brine with constant stirring for 1 min as pre-wash and homogenisation. Randomised portions were then divided into 6 groups, and excessive fluid was drained by placing grouped portions without overlapping on a clean rack for 30 s prior to sanitisation treatments. Integra Twin Oxide (ClO_2_) (Yangebup, WA, Australia) and PureSan hypochlorous acid (HOCl) (Sebenza, SA, Australia) were supplied by Integra Water Treatment Solution (Yangebup, WA, Australia). Ethanol, buffered formalin, magnesium oxide (MgO), boric acid, methylene blue, methyl red, Alcian blue, hydrochloric acid (HCl), trichloroacetic acid (TCA), 2,6-di-tert-butil-4-methylphenol (BHT), 2-thiobarbituric acid (TBA), 1,1,3,3-tetrathoxypropane (TEP), and other consumables were purchased from Sigma-Aldrich Pty Ltd. (Macquarie Park, NSW, Australia) and Thermo Fisher Scientific Australia Pty Ltd. (Scoresby, VIC, Australia). Plate count agars were supplied by Merieux NutriSciences (Perth, WA, Australia).

### 2.2. Treatments and Study Setting

Grouped fish portions were dropped into either sterile brine solutions for control and ClO_2_ ice treatments or dropped into freshly diluted sanitisers with 5 ppm ClO_2_, 10 ppm ClO_2_, 50 ppm HOCl, or 100 ppm HOCl per the dosage recommendation of the manufacturer. The pH of the brine and diluted sanitisers was recorded prior to sanitisation (Table 1). After dropping the samples into the solutions, continuous stirring was conducted for 5 min and the samples were then drained for 30 s. Fish portions were then placed on sterilised ice inside polystyrene boxes for all treatments except the ClO_2_ ice treatment, which was placed on ClO_2_-infused sterilised ice inside a polystyrene box. Uncovered polystyrene boxes were then placed in a simulated retail display refrigerator, which was maintained at 4 °C. Melted ice was drained, with new ice replaced daily. Physical, chemical, and microbiological analyses were then conducted on days 1, 4, and 7 post-sanitisation.

### 2.3. Physical Analyses

#### 2.3.1. Drip Loss

The modified method of Zhu et al. [33] was used to measure drip loss. Fish portions were weighed individually after grouping on the treatment day and days 1, 4, or 7. Three fish portions were measured on each analysis day per treatment, and drip loss was then computed using the following equation:(1)$$\mathrm{Drip}\;\mathrm{loss}\;\left(\%\right)=\frac{\mathrm{Initial}\;\mathrm{weight}\;\mathrm{of}\;\mathrm{fish}\;\mathrm{portion}\;\left(g\right)-\mathrm{Current}\;\mathrm{weight}\;\mathrm{of}\;\mathrm{fish}\;\mathrm{portion}\;\left(g\right)}{\mathrm{Initial}\;\mathrm{weight}\;\mathrm{of}\;\mathrm{fish}\;\mathrm{portion}\;\left(g\right)}\times 100\%$$

#### 2.3.2. Quality Index Method (QIM)

The quality index (QI) on Atlantic Salmon (*Salmo salar*) developed by Fuentes-Amaya et al. [34] was used to the determine the sensory quality of fish portions. On each sampling day, two trained seafood researchers assessed the quality of individual fish fillets (flesh, skin, and overall appearance) independently. Aspects including colour, translucency, texture, gapping, and odour were assessed based on a validated scoring system. Scores were then summed out of thirteen, with a high demerit score indicating greater deterioration in quality. Triplicates were conducted per treatment on each day.

#### 2.3.3. Texture Profile Analysis (TPA)

Along the lateral line of the fish portions, two 2.5 × 2.5 × 2.5 cm (width × length × height) fish cubes were isolated and tempered at room temperature for 30 min. Texture analyses were then conducted using a texture analyser TVT 6700 (PerkinElmer Inc., Waltham, MA, USA) with TexCal texture analyser software (version 5.0). The instrument was equipped with a 20 kg load cell and a 25 mm diameter stainless-steel flat-ended cylindrical probe. The test condition was set to be two consecutive cycles with 50 % compression under a constant speed of 50 mm/min and 5 sec rest time between cycles. Per each fish cube, six texture parameters were determined, including hardness (g), cohesiveness (ratio), adhesiveness (g/s), springiness (mm), gumminess (g), and chewiness (g/mm). Mathematical formulas for the calculation of each parameter were based on Bourne [35]. Two measurements were obtained from each fish portion and triplicate portions were prepared per treatment each day.

#### 2.3.4. Colour

Surface colour coordination (L*, a*, b*) was obtained using HunterLab ColorFlex (Hunter Association Laboratory Inc., Reston, VA, USA) with a glass port insert. Colour measurements were taken in triplicate at the dorsal and ventral locations of flesh and skin adjacent to the lateral line. Three portions were measured per treatment per day, and C* was calculated using the formula listed below:(2)$$C*=\sqrt{{a*}^{2}+{b*}^{2}}$$

#### 2.3.5. Histology

An approximately 1 mm-thick fillet tissue 2.5 cm away from the lateral line in the dorsal muscle of the fish portion was sampled. Triplicate fillet samples per treatment were fixed immediately in 10% buffered formalin after cutting with a sharp knife. Dehydration was then conducted through an alcohol series. After that, samples were embedded in paraffin, cut to 5 µm thickness, and stained with Alcian blue. The observation was then conducted under a light microscope according to the standard histological procedure.

### 2.4. Chemical Analyses

#### 2.4.1. Thiobarbituric Acid Reactive Substance (TBARS)

Lipid oxidation was measured using a modified 2-thiobarbuituric acid reactive substance (TBARS) assay by Cardoso et al. [36]. Two g of samples were blended with 8 mL of 5% TCA along with 0.2 mL 0.15% BHT in ethanol using an IKA-T18 homogeniser (IKA-Werke GmbH & Co., KG, Staufen, Freiburg, Germany) at 13,500 rpm for 15 s. Homogenates were then filtered through Whatman no. 6 filter paper, collected, and adjusted to 10 mL using 5 % TCA. To reduce interference due to turbidity, the filtrate was pumped through a 0.45 µM disc filter under pressure. One mL of clarified sample was transferred to a screw cap test tube with 1 mL of 0.08 M TBA added and boiled for 10 min at 100 °C. A total of 200 µL of solution were transferred to a 96-well plate, and absorbance was determined at 532 nm. A standard curve was prepared using 0 to 10 µM TEP in 20% TCA. Duplicates from three fillet portions were conducted per treatment each day.

#### 2.4.2. Total Volatile Base Nitrogen (TVB-N)

The method mentioned by Chung et al. [37] based on Zhang et al. [38] and Özoğul and Özoğul [39] was used to estimate the TVB-N of samples. Four g of fish samples were mixed with 40 mL of deionised water and homogenised using an IKA-T18 homogeniser (IKA-Werke GmbH & Co., KG, Staufen, Freiburg, Germany) at 13,500 rpm for 15 s. The mixture was then filtered using Whatman no. 1 filter paper, and filtrates were collected afterward. After that, 5 mL of filtrate were transferred to a digested tube and 5 mL of MgO (10 g L^−1^) were added immediately before commencing distillation. Distillation was conducted using a FOSS Kjeltec 2100 distillation unit (FOSS Pty Ltd., Hilleroed, Denmark, Europe), and the final product was collected in 10 mL boric acid with methylene blue and methyl red indicators. Distillation was deemed to be completed when it reached a 150 mL final volume. Distillates were titrated with 0.01 mol L^−1^ HCl to determine nitrogen content. Duplicate measurements from three fillet portions were conducted each day per treatment and TVB-N was calculated using the equation listed below:(3)$$T\mathrm{VB}-N\;(\mathrm{mg}/100\;g)=\frac{\mathrm{Titrated}\;\mathrm{volume}\;\left(\mathrm{mL}\right)\;\times \;14.01\;\times \;0.01\;\times \;(40+\mathrm{sample}\;\mathrm{weight}\;\left(g\right))\;\times \;100}{5\;\times \;\mathrm{sample}\;\mathrm{weight}\;(g)}$$

#### 2.4.3. pH

The method of Chung, Howieson, and Chaklader [37] and the Association of Official Agricultural Chemists [40] was used in this study to determine pH. In detail, minced fish samples were mixed with deionised water in a 1-to-10 ratio and homogenised using an IKA-T18 homogeniser (IKA-Werke GmbH & Co., KG, Staufen, Freiburg, Germany) at 13,500 rpm for 15 s. Samples were then mixed at 60 rpm for 30 min at 24.5 °C. After that, pH was measured using a three-scale calibrated Aqua-pH meter (TPS Pty Ltd., Berndale, QLD, Australia). Measurements were carried out in duplicate with triplicate sampling per treatment each day.

### 2.5. Microbiological Analyses

#### Total Viable Count (TVC)

The AS 5013.5 standard method [41] was used to measure the total viable count of fish portions. The fish portion was homogenised and diluted, and 1 mL of homogenate was added to a sterile Petri dish. Plate count agar that was heated to 44 °C was then poured on top. The inoculum was then mixed by rotating Petri dishes and left to solidify. Incubation was conducted at 30 °C for 72 h, and colonies were counted afterward. Data was then expressed as log CFU/g.

### 2.6. Statistical Analyses

SPSS version 27 (IBM Australia Ltd., St Leonards, NSW, Australia) was used to perform all statistical analyses. Two-way analysis of variance (ANOVA) with treatments and storage times as main factors was conducted on all physical and chemical parameters. Simple main effect analyses were conducted afterwards with subsequent one-way ANOVA and Tukey’s post-hoc test after confirmation of ANOVA assumption through Shapiro–Wilk and Levene’s tests. Significant level was set at *p* < 0.05 and results are presented as mean ± standard deviation (SD). Linear regression models were also conducted to identify changes in trends and results are presented as Pearson’s correlation coefficient (r) and unstandardised regression coefficient (B).

## 3. Results and Discussion

### 3.1. Physical Analyses

#### 3.1.1. Drip Loss

Drip loss is of importance in salmon production because it measures the amount of fluid released from muscle products, which are directly related to sensory loss, including texture and flavour; nutrient loss, such as protein and amino acids; water-holding capacity, and financial implications of salmon or salmon products [42,43,44]. Whilst according to Figure 1A there was no significant difference between the control and all sanitisers on days 1 and 7, during day 4, drip loss was higher in the control than in both HOCl treatments (*p* = 0.019; 0.004). Thus, to allow for a clear understanding of the overall impacts of the sanitiser, Figure 1B was generated, with all treatments found to insignificantly reduce drip loss except HOCl at a 100 ppm dosage, which reduced drip loss significantly compared to control-treated fillets. A similar drip loss-reducing effect has previously been noted in beef treated with HOCl [45], suggesting that HOCl treatment could be ideal for maintaining sensory quality and potential production loss during meat processing [42,43,44]. A similar relationship between drip loss and sensory-related parameters were noted in the current study as well, with a significant correlation between sensory-related parameters (QI, b*, C*, TVB-N, and pH) observed with drip loss (Figure 2). This indicates that the drip loss protection effect from HOCl could assist in maintaining the quality of fish during storage. Although this phenomenon has not been well studied in the context of seafood, a previous literature review on the drip loss mechanism in pork by Huff-Lonergan and Lonergan [46] suggested that early-stage degradation of specific muscle protein or structure prior to storage could reduce the physical constraints that limit water-holding capacity, resulting in improved fluid retention. Early-stage degradation was also found in this study (Figure 3B–F), reinforcing the possibility of the proposed theory—a reduction in drip loss due to initial treatment-induced damage. Similarly, storage time impacted the drip loss and showed a significant increase with prolonged storage time for all treatments (Figure 1A), which was further supported by the strong positive correlation between drip loss and storage day among all treatments (r ≤ 0.79). The results align with trends in other studies that investigated the impact of storage on drip loss. The mechanisms have been discussed in great depth by Chan et al. [47]. It is as well noted that the drip loss on day 1 for all treatment was below 2%, which was within the acceptable limit. However, the drip loss on days 4 and 7 for all treatments exceeded the limit irrespective of treatment [48,49]. To summarise, as evidenced by days 4 and 7, drip loss could be reduced by treatments during shelf life. Yet, the degree of reduction might not be sufficient to be detected as an abatement of deterioration during storage.

#### 3.1.2. Quality Index Method (QIM)

QIM is a standard and reliable tool for assessing freshness and quality by measuring the degree and rate of change in fishery products [50,51]. The scores of validated quality-related parameters are summarised to provide the quality index, which increased linearly with storage time on ice [52]. Similarly, the QI score increased linearly with increasing storage time (Figure 4), and the linearity was greatly contributed to by the highest demerit scores in skin brightness, flesh odour, and gaping. A similar skin-lightening effect induced by the sanitisers was as well reflected in colorimetric data (Table 2). Flesh odour was also supported by rancidity production (MDA) in sanitiser-treated fillets, and textural changes were reflected in significant correlation with springiness and adhesiveness (Figure 2). The linearity was expected in the present trial since many studies report a linear relationship between QI and storage time in finfish [53,54,55,56]. In terms of regression models, CIO_2_ ice, CIO_2_ at 10 ppm, and HOCl at 100 ppm were effective in retarding the increase in QI over time, as evidenced by the unstandardised regression coefficient (B), which was found to be lower than the control. However, at lower dosages, CIO_2_ and HOCl either worsen or were shown to have no impact on QI increase over time. It is also worth noting that, at 5 ppm, CIO_2_ was more effective as an ice infusion in comparison to a dip solution, which concurs with a previous study by Shin et al. [57]. Those authors hypothesised that antimicrobial ice is a more optimal treatment due to its ability to sustain release antimicrobial agent and keep the microbial population at a lower level throughout the shelf life. Our results indicated that to reduce physical quality deterioration over time in salmon, CIO_2_ ice, CIO_2_ at 10 ppm, or HOCl at 100 ppm could be utilised.

#### 3.1.3. Texture and Colour Profiles

Texture profiles are frequently applied in relation to other parameters to examine and evaluate fish physical quality alteration during shelf-life studies [58]. Overall, texture alteration was protected by sanitiser treatments during storage, with no significant differences in textural changes observed in all parameters until the end of the trial (Table 2). However, on the contrary, the control was found to have texture deterioration, as evidenced by the cohesiveness, gumminess, and hardness changing significantly by day 7 (Table 2). The significant textural changes in control fillets, especially in hardness on day 7 when compared with days 1 and 4, are in alignment with TVBN production (Figure 5), suggesting that the texture changes observed in the current study could be related to advance protein breakdown during storage time. Similar textural protective effects of treatments were noted in a previous study by Xu et al. [59] and Lan et al. [60], which identified protection effects on texture changes when HOCl or ClO_2_ were used to disinfect *Alosa sapidissima* and *Pseudosciaena crocea*. In Lan, Zhou, Zhao, and Xie’s [60] study, the hypothesis of HOCl or ClO_2_ textural protection due to a delay in protein alteration was confirmed, with various analyses demonstrating enhancement in the structural stability of fish protein and a reduction in activities of endogenous proteases over time, suggesting that chlorine-based treatments could be effective in protecting against texture changes during storage.

The consumer purchasing decision regarding salmon fillets is largely influenced by colour, which can change during storage. The lightness (L*) of fillets was not affected by sanitisers but was influenced by storage time. Lightness in the control, ClO_2_ ice (5 ppm)-, and ClO_2_ (5 ppm)-treated fillets increased on day 7 compared with days 1 and 4. Such increases were not noted in ClO_2_ (10 ppm) or HOCl (50, 100 ppm), demonstrating that perhaps higher dosages of CIO_2_ or HOCl treatment were more appropriate in protecting against a lightness increase during salmon fillet storage, possibly acting by reducing the increase in opaqueness due to protein denaturation [37,61,62].

Meanwhile, skin lightness decreased significantly in all sanitiser-treated salmon on all days. The same skin discoloration effects induced by chlorine were as also developed in red grouper (*Epinephelus morio*) and salmon treated with 100 or 200 ppm ClO_2_ in a previous study [31]. Since dark pigmentation on the skin of salmonids is melanin-based, the intrinsic nature of high reactivity between melanin and chlorine-based disinfectants could possibly explain the current observation—the bleaching of melanin pigments, which result in skin lightening [63,64,65]. Additionally, there was a loss of skin lightness over the storage time irrespective of sanitiser application, suggesting that sanitisation was not effective in controlling the colour of salmon skin. The changes in lightness over time have been suggested to be associated with lipid oxidation and microbial spoilage, although weak correlations were found with TBARS and TVC values in the current study (Figure 2) [37,66,67]. Yet, it is inconclusive whether such effects are truly undesirable due to the lack of research conducted on consumers’ colour preference of fish skins.

The redness colour coordination (a*) is commonly used to evaluate the freshness of salmon, with a high a* generally being more preferrable by consumers and a value below 13 being unacceptable [61]. There were no differences in a* between treatments until day 4, suggesting that sanitisation had no direct effect on the initial a* value of salmon fillets. Meanwhile, the a* in all salmon fillets, especially in ClO2 (10 ppm)- and HOCl (50 ppm)-treated samples, decreased over the storage time, which was mainly due to brown-coloured substances formed by oxidation, which are enzymatic or non-enzymatic browning reactions [37,66]. Regardless of sanitisers, skin redness increased gradually with increasing storage time, which could be a consequence of undergoing similar reactions as salmon flesh.

It has been reported that deterioration results in yellower fish meat over the storage time [68,69] which is related to similar reactions that affected redness (a*). Similarly, on day 7, fillet yellowness increased significantly in fillets treated with sanitisers compared with the control. Xiong et al. [70] also found an increase in yellowness in salmon fillets treated with salmon bone gelatine with chitosan, gallic acid, and clove oil during 15 days of cold storage time. The skin yellowness over the 7 days of storage time increased gradually, though sanitisers did not affect skin yellowness.

Chroma (C*), on the other hand, represented the overall intensity of colour, which has been suggested to be one of the main factors that affect consumer desire to purchase salmon. A deep orange or red colour, or high C*, is considered to be desirable by consumers across different demographics [71]. While C* of the control decreased over time, possibly due to oxidation of pigmented antioxidants such as astaxanthin and carotenoid, which are strongly associated with salmon colour [72], on day 7, CIO_2_ and HOCl dip treatments were shown to have a higher C* than the control. Although the differences were not significant, dip treatments included in this study might have had a protective effect on colour degradation.

#### 3.1.4. Changes in Muscle Microstructure

The microstructure of muscles showed that there was a reduced fibre–fibre adhesion and an increase in interfibrillar space on day 1 in all treatments (Figure 3B–F) compared to the control, while ClO_2_ ice (5 ppm) seemed to preserve adhesion more on days 4 and 7 (Figure 3H,N). The initial increase in TVBN in all treatments on day 1 compared to the control was also most likely a result of cell damage found in histology. However, as discussed in the previous section, this might be ideal in drip loss reduction over storage time. Low TPC count could mostly be linked to minimal myofibrilla damage, as indicated in the control, while the damage of other treatments led to an increase in nutrients available for microbes. The increase in space was possibly due to fluid build-up between muscle fibre and the shrinkage of cells. An increase in intrafibrillar cavities was found in ClO_2_ (10 ppm) (Figure 3D), HOCl (50 ppm) (Figure 3E), and HOCl (100 ppm) (Figure 3F) on day 1. An increase in space between myofibrilla and more foci (infected tissue) was found on days 4 and 7 of HOCl treatment (Figure 3K,L,Q,R) and in ClO_2_ (10 ppm) (Figure 3P) on day 7. Increased storage time increased interfibrillar space due to shrinkage of muscle in all treatments except ClO_2_ ice (5 ppm), which preserved the adhesion. ClO_2_ ice (5 ppm) (Figure 3B,H,N), ClO_2_ (10 ppm) (Figure 3D,J,P), and HOCl (100 ppm) seemed to reduce severity, especially on day 7. There was an increase in the amount of intrafibrillar damage over time, especially in ClO_2_ (5 ppm dip) and HOCl (100 ppm) treatments. Low-dosage HOCl led to severe cell shrinkage, while a higher dosage led to the formation of necrotic tissue, indicated as light-coloured foci. The histological differences found in treatments in comparison to the control could explain the protective effects against gumminess, chewiness, and hardness identified during storage in the texture analysis. In addition, tenderising effects caused by chlorine-based treatments could be related to the microstructural and cytological damage identified above.

### 3.2. Chemical Analyses

#### 3.2.1. Thiobarbituric Acid Reactive Substance (TBARS)

Fish, especially fatty fish, are prone to oxidation because of the presence of lipids. Hence, the quantification of lipid secondary oxidation substances, in particular, malondialdehyde (MDA), measured by TBARS analysis is widely used to evaluate the lipid oxidation products [73,74]. Two-way analysis showed a significant effect of sanitiser and storage time, with an interaction on the MDA production level (Figure 5). Sanitiser-treated fillets of salmon showed a significantly higher level of MDA compared with the control. When data across individual days were pooled, an increase in lipid oxidation was observed, as shown in Figure 5B. HOCl negatively impacted the MDA, and fillets treated with 100 ppm HOCL had the highest MDA levels on all days compared to the control (Figure 5B). Similar oxidation accelerating effects by chlorine sanitisers were noted in a previous study on salmon fillets [20]. The current finding was not of surprise, since chlorine sanitisation works via oxidation; thus, it may non-selectively oxidise cell membrane fatty acid, which leads to an increases MDA [75,76]. In fact, in Kim, Lee, O’Keefe, and Wei’s [20] study, a dose-related increase in lipid oxidation was identified. In terms of the impact of storage time, a strong positive relationship with MDA was found (Figure 5A). Similarly, in a previous study, MDA increased significantly in salmon fillets when treated with a freshly prepared aqueous solution of ClO_2_ at 20, 40, 100, and 200 ppm for 5 min [20]. Irrespective of sanitiser and storage time, MDA production exceeded the acceptable level of MDA (1–2 mg MDA/kg fish) in salmon fillets over time [70], implying that all the samples suffered a high degree of lipid oxidation. Yet, it is worth noting that the CIO_2_ ice treatment has a lower MDA level in comparison to the dip treatment (Figure 5B), suggesting that perhaps exploration of the effects of various dosages of ice infused with chlorine sanitiser could be the ideal next step.

#### 3.2.2. Total Volatile Base Nitrogen (TVB-N)

The decomposition of fish proteins by endogenous enzymes and spoilage bacteria during storage generates TVB-N production, which is composed of ammonia, trimethylamine oxide, trimethylamine, and dimethylamine [77]. There were no significant differences in all treatments on days 4 and 7 (Figure 5C). On day 1, the control had significantly lower TVB-N; however, it was the highest after that even though the differences were not significant. Insignificant differences were found between all days besides a significant elevation in TVB-N in the control at day 7. However, an increase in TVB-N was observed with time (Figure 5C). A significant interaction was found, contributed to by the control only and on day 7 only (Figure 5C). The TVB-N increased from 6.07 on day 1 to 21.65 mg/100 g in salmon fillets, which is within the acceptable limits (25-40 mg/100 g) [77,78], suggesting that sanitiser might have prevented protein oxidation, aligning with the discussion above. Similarly, TVB-N was below the critical limit until 12 days of storage time but exceeded the threshold limit after 15 days in salmon fillets treated with the control and chitosan-based coatings [79].

#### 3.2.3. pH

On days 1 and 4, pH did not differ among all treatments, whilst ClO_2_ ice-treated fillets showed a slight elevation in pH, followed by ClO_2_ (5 ppm) and control on day 7 (Figure 6A). The increase in pH in ClO_2_-treated salmon fillets contradicted with the findings of Kim, Huang, Marshall, and Wei [31], who reported lower levels of pH in salmon fillets treated with 20–100 ppm, suggesting dose-dependent effects of ClO_2_ on pH. Yet, the contradicting results could possibly be a consequence of the relatively low dosage utilised in the current study (<10 ppm). The increased pH aligned with bacterial load in fillets treated with the same sanitiser. Both HOCl treatments had a significantly lower pH than the control, with 100 ppm having the lowest, which was not aligned with bacterial data. A positive r value was recorded for all days, meaning that storage time positively correlated with pH value. Irrespective of sanitiser effects, the pH increase over the storage time (Figure 6A) was consistent with the findings of the total plate count, MDA levels, and, mostly importantly, TVBN, which has been shown to have a direct correlation with pH increase during the storage of salmonids [80,81]. The changes in pH from 6.07 on day 1 to 6.55 on day 7 were within the normal range of fresh salmon pH [82], indicating minimal production of non-desirable alkaline compounds derived from microbial activity, which is evidenced by moderate correlation between pH and TVB-N (Figure 2) [83]. This change was aligned with the production of acceptable levels of TVBN in salmon fillets during 7 days of storage.

### 3.3. Microbiological Analyses

TVC is an index commonly used to investigate the microbial quality of food, and the maximum acceptable limit for quality and freshness of fish products is generally a value of 7 log CFU/g [67]. The TVC value on days 1 and 4 being below 6.00 log CFU/g increased in salmon fillets treated with sanitisers in comparison with control fillets but remained below the acceptable limit (Figure 7). This may suggest that both ClO_2_ and HOCl can prevent the proliferation of spoilage bacteria until day 4. A similar effect of 15 and 50% of HOCl was observed in the study of Khazandi, Deo, Ferro, Venter, Pi, Crabb, Amorico, Ogunniyi, and Trott [32], who reported TVC below the acceptable limit until day 3 in King George Whiting (*Sillaginodes punctatus*) fish fillets, and repeated wash (10 s + 5 min) decreased overall bacterial loads in salmon dipped in either 15 or 50 % of HOCl solution. Additionally, immersion of various fish products in electrolysed water solution and ice containing HOCl has been reported to produce anti-microbial activity in various studies, summarised in the review by Ramírez Orejel and Cano-Buendía [84]. Meanwhile, in the current study, salmon fillets treated with ClO_2_ ice (5 ppm), ClO_2_ (10 ppm), and HOCl (100 ppm) exceeded the permissible limit of the TVC value on day 7. The low TPC count in the control fillets could mostly be linked to minimal myofibrilla damage, as indicated in the control, while the damage in other treatments led to an increase in cell lysis, resulting in increasing nutrients available for microbes. The changes in TVC on days 1 and 4 were consistent with changes of TVC in whole salmon muscle on days 1 and 3 [31] but exceeded the limit of TVC on day 7 in the present study. This was not consistent with the findings of Kim, Huang, Marshall, and Wei [31], who reported TVC values within the acceptable limit. However, keeping dosage and method differences in mind, the negative results in the current study could have been due to the usage of a relatively low dosage and the current dipping method. In a previous study by Khazandi, Deo, Ferro, Venter, Pi, Crabb, Amorico, Ogunniyi, and Trott [32], it was suggested that bacterial load could be maintained within the acceptable range following a single 5 min wash with chlorine-based sanitisers. Hence, further study is needed to optimise sanitisation methods, including factors such as dipping methods, the introduction format (e.g., ice, wash or dip), and the usage of a larger range of sanitiser dosage to extend the shelf life of Atlantic salmon.

## 4. Conclusions

All sanitiser treatment reduced drip loss on in this study, although a significant reduction was only observed in the HOCl treatment at the 100 ppm dosage. A strong relationship was found between the total QI score and storage time in all treatments, which was mainly influenced by the highest demerit score in skin brightness, flesh odour, and gaping. Texture parameters such as gumminess, chewiness, and hardness alternation over time was protected by sanitisers. Tenderising effects of fillets could be related to structural and cytological damage revealed in the fillet muscle following histological observation. Though MDA levels exceeded threshold levels during the storage time, TVB-N was within permissible levels during the storage time. Similar to TVB-N, pH fell within the normal range for fresh salmon. TVC was below the permissible levels until day 4 but exceeded the threshold levels in ClO_2_ ice (5 ppm), ClO_2_ (10 ppm), and HOCl (100 ppm) on day 7. However, sanitisers were effective in controlling the degradation of protein, which was supported by TVB-N results and microbial spoilage until day 4. The mixed results suggest that further study is needed, especially in consideration of specific spoilage bacteria and the treatment effect on fish that are less susceptible to lipid oxidation.

## Figures and Tables

**Figure 1 foods-13-03156-f001:**
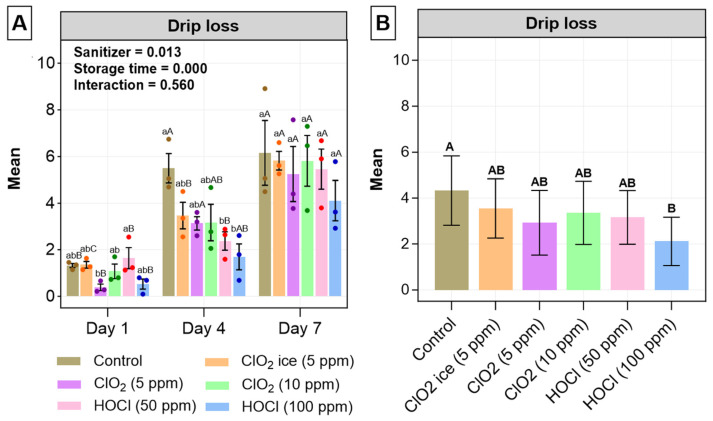
Drip loss (%) changes in salmon fillets for 7 days post-sanitisation. Values are expressed as mean ± standard deviation from three biological replicates per day. (**A**) Drip loss (%) of all treatments during the 7-day study. Significant differences between the treatments within individual days are indicated with lowercase letters, whereas significant differences in each treatment across storage time are indicated with uppercase letters. (**B**) Pooled data of treatments to identify the mass effect of sanitiser on drip loss. Significant differences are labelled with uppercase letters. The level of significance is set at *p* < 0.05. the *p*-values of storage time, treatment, and interaction are obtained from a two-way ANOVA test.

**Figure 2 foods-13-03156-f002:**
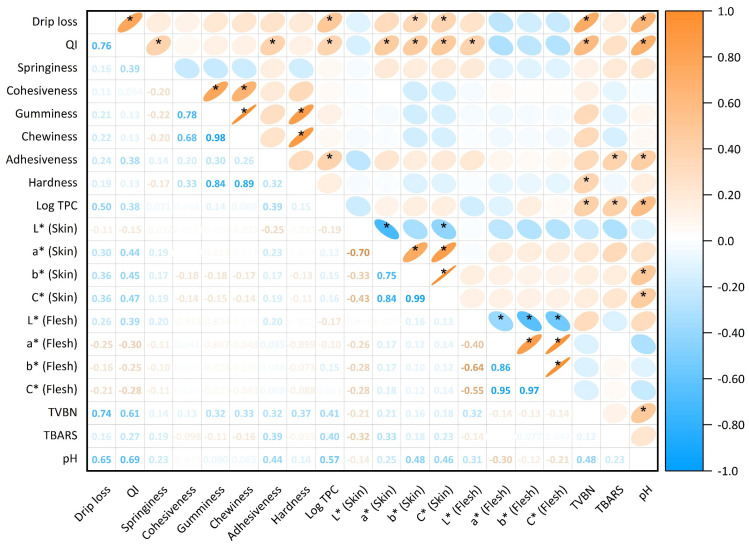
Pearson’s correlation matrix of all investigated parameters in the current study. Significant correlations are labelled with (*). For visualisation purposes, positive correlations are coloured in orange, while negative correlations are represented in blue. The intensity of the colour and a reduction in the size of the ellipse indicate the significance of the relationship.

**Figure 3 foods-13-03156-f003:**
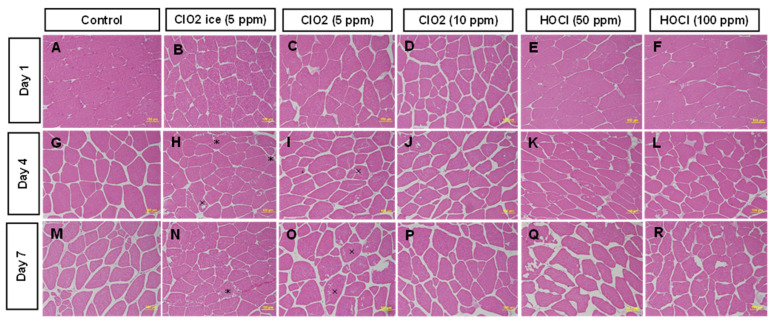
Transversal section of muscle tissue of salmon fillets for 7 days post-sanitisation. Figure was labelled from (**A**–**R**) according to treatments and storage time. (*) Indicates loss of integrity of sarcolemma, characterised by indistinct margins; (×) indicates intrafibrillar cavities, characterised by spacing between myofibrilla within muscle fibre.

**Figure 4 foods-13-03156-f004:**
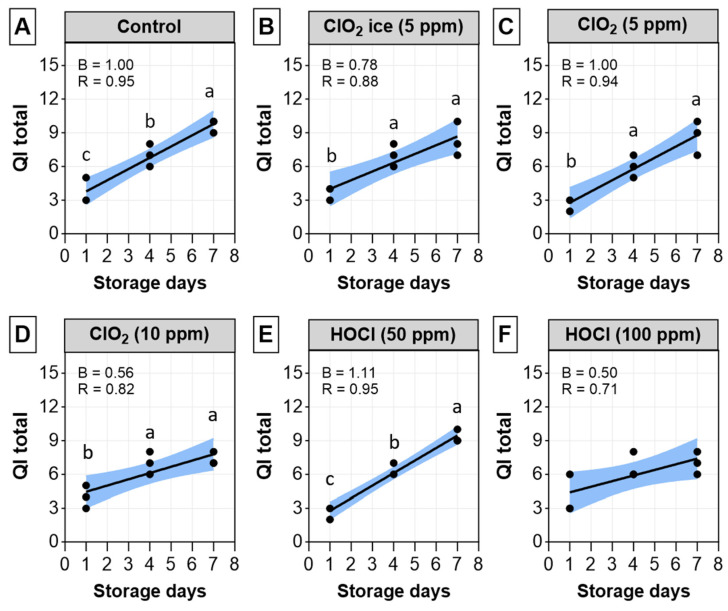
Quality index (QI) changes in salmon fillets for 7 days post-sanitisation. (**A**) Control; (**B**) ClO_2_ ice (5 ppm); (**C**) ClO_2_ (5 ppm); (**D**) ClO_2_ (10 ppm); (**E**) HOCl (50 ppm); (**F**) HOCl (100 ppm). The relationship between storage day and drip loss in individual treatments is expressed as the Pearson’s correlation coefficient (r) and unstandardised regression coefficient (**B**). Significant differences within treatments across different days are indicated with lowercase letters. No significant differences were detected when comparing individual treatments on each sampling day.

**Figure 5 foods-13-03156-f005:**
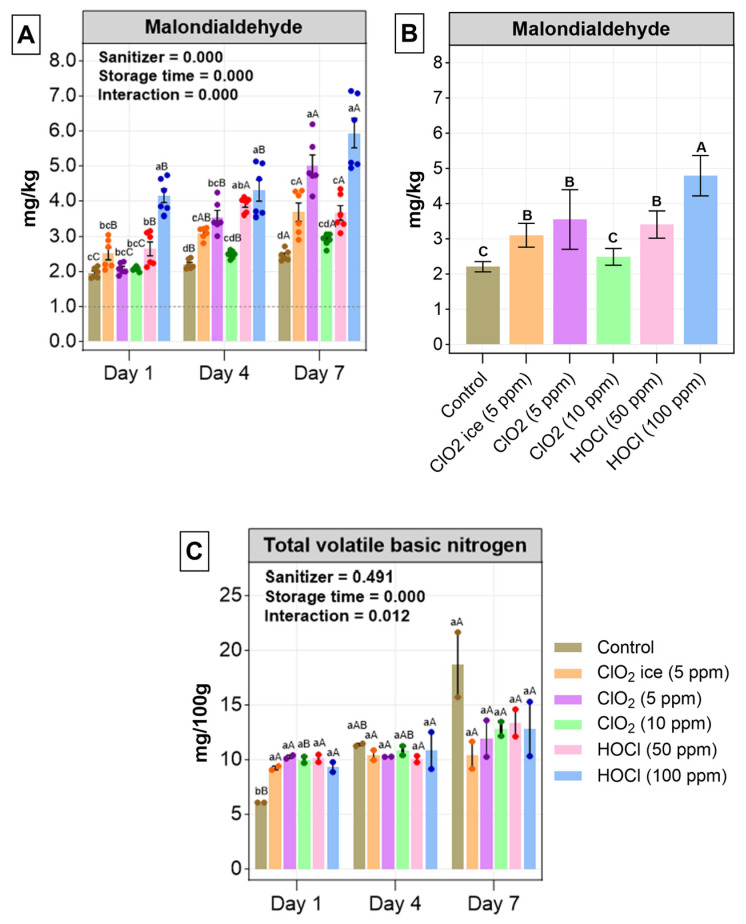
Lipid oxidation (**A**) and total volatile base nitrogen (TVB-N) (**C**) across treatments during the 7-day study. (**B**) General lipid oxidation across treatment. Values are expressed as mean ± standard deviation from three biological replicates per day. In (**A**,**C**), uppercase letters indicate significant differences between storage for individual treatment, and lowercase letters indicate significant differences between treatments during each sampling day. In (**B**), significant differences between treatments are indicated with uppercase letters. The level of significant is set at *p* < 0.05. *p*-Values of storage time, treatment, and interaction were obtained with a two-way ANOVA test.

**Figure 6 foods-13-03156-f006:**
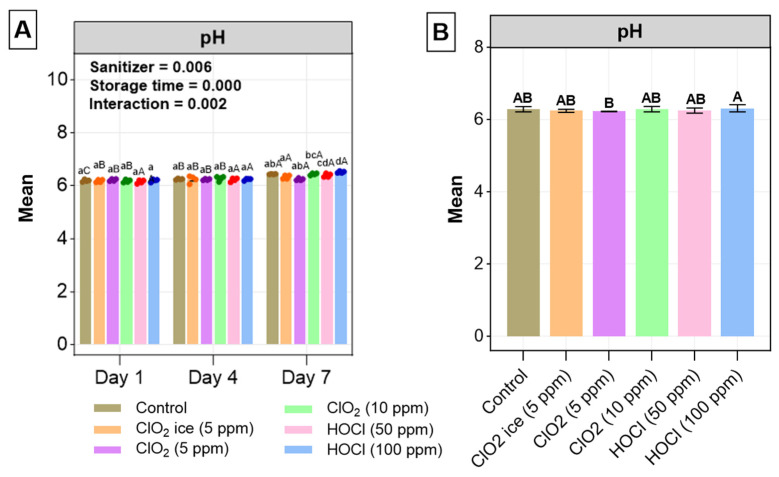
(**A**) pH changes in salmon fillets for 7 days post-sanitisation. (**B**) General effects of treatments on pH. Values are expressed as mean ± standard deviation from three biological replicates per day. In (**A**), the uppercase letters indicate significant differences between storage for individual treatment and the lowercase letters indicate significant differences between treatment during each sampling day. In (**B**), significant differences between treatments are indicated with different uppercase letters. The level of significance is set at *p* < 0.05. *p*-Values of storage time, treatment, and interaction were obtained from two-way ANOVA test.

**Figure 7 foods-13-03156-f007:**
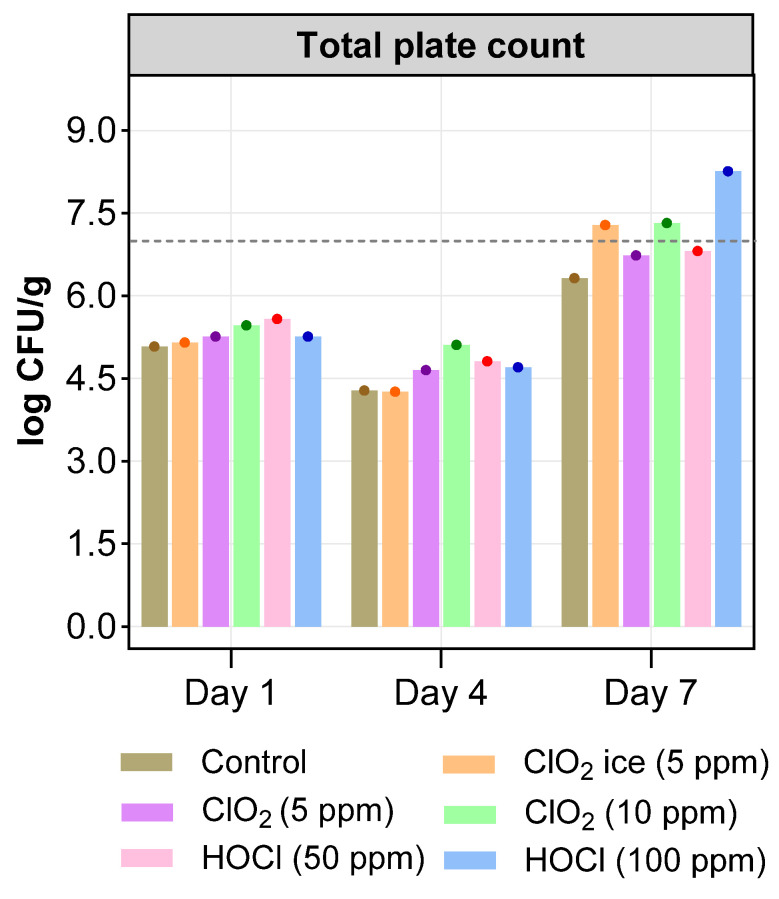
Total viable count (TVC) of salmon fillets for 7 days post-sanitisation. Acceptable limit (log 7 CFU/g) was represented with dashed line (---).

**Table 1 foods-13-03156-t001:** pH of brine and diluted sanitisers.

Solution	pH
3.5% brine	7.64 ± 0.09 ^a^
5 ppm ClO_2_	7.18 ± 0.01 ^c^
10 ppm ClO_2_	6.84 ± 0.11 ^d^
50 ppm HOCl	7.50 ± 0.04 ^ab^
100 ppm HOCl	7.35 ± 0.01 ^bc^

Recording was conducted under room temperature at 24.5 °C. Values are expressed as means ± SD of duplicate measurements. Different lowercase letters indicate significant differences between treatments (*p* < 0.05).

**Table 2 foods-13-03156-t002:** Changes in texture profile and colour in salmon fillets for 7 days post-sanitisation.

	Treatments						Two-Way ANOVA (P)		
	Control	ClO_2_ Ice (5 ppm)	ClO_2_ (5 ppm)	ClO_2_ (10 ppm)	HOCl (50 ppm)	HOCl (100 ppm)	T	ST	T × ST
Texture Profile									
Springiness (mm)							0.419	0.139	0.673
Day 1	1.00 ± 0.00 ^aA^	1.00 ± 0.01 ^aA^	0.99 ± 0.03 ^aA^	1.00 ± 0.00 ^aA^	0.99 ± 0.03 ^aA^	1.00 ± 0.00 ^aA^			
Day 4	1.00 ± 0.00 ^aA^	1.00 ± 0.00 ^aA^	1.00 ± 0.00 ^aA^	1.00 ± 0.00 ^aA^	1.00 ± 0.00 ^aA^	1.00 ± 0.00 ^aA^			
Day 7	0.99 ± 0.02 ^aA^	0.99 ± 0.02 ^aA^	1.00 ± 0.01 ^aA^	1.00 ± 0.00 ^aA^	1.00 ± 0.00 ^aA^	1.00 ± 0.00 ^aA^			
^1^ B (r)	−8.24 × 10^−4^ (−0.21)	−5.56 × 10^−4^ (−0.10)	1.94 × 10^−3^ (0.31)	-	1.62 × 10^−3^ (0.31)	-			
Cohesiveness (ratio)							0.066	0.008 *	0.897
Day 1	0.23 ± 0.01 ^aB^	0.26 ± 0.04 ^aA^	0.25 ± 0.03 ^aA^	0.26 ± 0.05 ^aA^	0.24 ± 0.11 ^aA^	0.20 ± 0.03 ^aA^			
Day 4	0.26 ± 0.05 ^aAB^	0.29 ± 0.03 ^aA^	0.26 ± 0.04 ^aA^	0.32 ± 0.08 ^aA^	0.27 ± 0.05 ^aA^	0.25 ± 0.05 ^aA^			
Day 7	0.27 ± 0.02 ^aA^	0.28 ± 0.06 ^aA^	0.26 ± 0.04 ^aA^	0.27 ± 0.04 ^aA^	0.27 ± 0.03 ^aA^	0.25 ± 0.03 ^aA^			
B (r)	6.58 × 10^−3^ (0.42)	3.06 × 10^−3^ (0.18)	2.50 × 10^−3^ (0.16)	4.27 × 10^−5^ (0.00)	4.02 × 10^−3^ (0.16)	9.00 × 10^−3^ (0.49)			
Gumminess (g)							0.310	0.022 *	0.486
Day 1	679.61 ± 211.84 ^aB^	1050.72 ± 473.39 ^aA^	898.04 ± 271.88 ^aA^	788.1625 ± 368.76 ^aA^	817.66 ± 513.72 ^aA^	609.58 ± 182.19 ^aA^			
Day 4	901.96 ± 256.29 ^aB^	1001.01 ± 329.93 ^aA^	826.83 ± 398.21 ^aA^	1079.79 ± 518.26 ^aA^	987.88 ± 293.71 ^aA^	789.01 ± 242.40 ^aA^			
Day 7	1307.61 ± 274.14 ^aA^	1034.23 ±413.41 ^aA^	950.76 ± 269.90 ^aA^	1067.77 ± 295.07 ^aA^	985.69 ± 291.78 ^aA^	927.82 ± 246.25 ^aA^			
B (r)	1.06 × 10^2^ (0.74)	−2.75 (−0.02)	8.79 (0.07)	4.01 × 10 (0.23)	2.58 × 10 (0.18)	5.30 × 10 (0.52)			
Chewiness (g/mm)							0.335	0.032 *	0.511
Day 1	771.38 ± 225.73 ^aB^	1115.04 ± 462.08 ^aA^	979.45 ± 295.81 ^aA^	848.50 ± 376.76 ^aA^	888.76 ± 440.32 ^aA^	707.67 ± 172.35 ^aA^			
Day 4	961.34 ± 240.34 ^aB^	1068.11 ± 331.35 ^aA^	894.82 ± 394.96 ^aA^	1100.87 ± 481.03 ^aA^	1035.35 ± 260.33 ^aA^	879.46 ± 250.71 ^aA^			
Day 7	1349.62 ± 260.64 ^aA^	1079.91 ± 368.58 ^aA^	1020.26 ± 208.57 ^aA^	1157.65 ± 250.71 ^aA^	1021.52 ± 290.82 ^aA^	968.77 ± 225.40 ^aA^			
B (r)	9.74 × 10 (0.72)	−5.85 (−0.04)	6.80 (0.05)	4.65 × 10 (0.29)	2.01 × 10 (0.16)	4.35 × 10 (0.46)			
Adhesiveness (g/s)							0.414	0.030 *	0.464
Day 1	−157.19 ± 97.85 ^aB^	−96.58 ± 61.15 ^aA^	−97.71 ± 42.21 ^aA^	−95.02 ± 35.44 ^aA^	−115.21 ± 39.79 ^aA^	−94.51 ± 50.14 ^aA^			
Day 4	−102.09 ± 55.87 ^aAB^	−120.43 ± 31.14 ^aA^	−97.85 ± 54.41 ^aA^	−113.91 ± 59.25 ^aA^	−94.73 ± 35.06 ^aA^	−67.58 ± 36.29 ^aA^			
Day 7	−56.10 ± 32.71 ^aA^	−96.23 ± 45.51 ^aA^	−55.61 ± 25.17 ^aA^	−114.02 ± 86.60 ^aA^	−63.71 ± 40.07 ^aA^	−65.88 ± 54.00 ^aA^			
B (r)	1.68 (0.56)	0.06 (0.00)	7.02 (0.39)	−2.93 (−0.11)	8.72 (0.51)	4.77 (0.26)			
Hardness (g)							0.735	0.006 *	0.168
Day 1	2941.80 ± 812.39 ^aB^	3888.33 ± 1221.31 ^aA^	3617.83 ± 796.22 ^aA^	2990.00 ± 831.56 ^aA^	3208.00 ± 690.23 ^aA^	3048.60 ± 482.02 ^aA^			
Day 4	3380.17 ± 515.34 ^aB^	3386.00 ± 859.30 ^aA^	3084.83 ± 1116.81 ^aA^	3487.83 ± 1039.67 ^aA^	3543.83 ± 601.23 ^aA^	3207.83 ± 723.00 ^aA^			
Day 7	4880.83 ± 1015.24 ^aA^	3613.00 ± 877.50 ^aA^	3632.00 ± 842.76 ^aA^	3956.00 ± 584.87 ^aA^	3660.83 ± 774.69 ^aA^	3633.40 ± 653.56 ^aA^			
B (r)	3.29 × 10^2^ (0.71)	−4.59 × 10 (0.12)	2.36 (0.01)	1.61 × 10^2^ (0.44)	7.27 × 10 (0.26)	9.75 × 10 (0.37)			
Colour (Flesh)									
L *							0.240	0.000 *	0.000 *
Day 1	51.97 ± 2.11 ^aB^	52.64 ± 2.46 ^aB^	52.48 ± 2.44 ^aB^	53.00 ± 1.08 ^aAB^	52.11 ± 2.22 ^aA^	52.72 ± 2.64 ^aB^			
Day 4	52.98 ± 1.31 ^cB^	55.13 ± 1.54 ^bcAB^	57.72 ± 3.47 ^abA^	55.99 ± 1.81 ^abcA^	53.56 ± 1.60 ^cA^	59.16 ± 3.14 ^aA^			
Day 7	58.17 ± 5.94 ^aA^	57.53 ± 4.92 ^aA^	54.60 ± 6.46 ^aAB^	52.23 ± 5.07 ^aB^	53.22 ± 2.66 ^aA^	52.03 ± 6.08 ^aB^			
B (r)	1.03 (0.58)	0.81 (0.54)	0.35 (0.18)	−0.13 (−0.09)	0.19 (0.21)	−0.12 (−0.05)			
a *							0.017 *	0.000 *	0.022 *
Day 1	25.20 ± 3.20 ^aA^	24.16 ± 1.63 ^aA^	26.13 ± 2.25 ^aA^	27.45 ± 0.90 ^aA^	26.76 ± 1.79 ^aA^	25.91 ± 2.56 ^aA^			
Day 4	24.21 ± 1.02 ^aA^	24.62 ± 1.36 ^aA^	24.07 ± 0.84 ^aA^	23.42 ± 1.98 ^aB^	23.64 ± 1.08 ^aB^	24.05 ± 1.06 ^aA^			
Day 7	23.11 ± 1.92 ^abA^	22.18 ± 4.08 ^bA^	24.29 ± 2.41 ^abA^	25.46 ± 1.70 ^abC^	23.83 ± 1.39 ^abB^	25.57 ± 2.09 ^aA^			
B (r)	−0.35 (−0.38)	−0.33 (−0.30)	−0.31 (−0.36)	−0.33 (−0.13)	−0.49 (−0.60)	−0.06 (−0.06)			
b *							0.001 *	0.000 *	0.002 *
Day 1	26.95 ± 2.82 ^abA^	25.64 ± 1.52 ^bA^	27.53 ± 1.74 ^abA^	29.00 ± 1.16 ^aA^	28.67 ± 2.76 ^aA^	27.47 ± 2.41 ^abA^			
Day 4	24.35 ± 1.00 ^aB^	25.10 ± 0.87 ^aA^	24.93 ± 1.00 ^aB^	24.54 ± 1.30 ^aB^	24.28 ± 0.69 ^aB^	24.23 ± 1.00 ^aB^			
Day 7	24.06 ± 1.28 ^cB^	24.66 ± 2.50 ^bcA^	26.98 ± 3.08 ^abcAB^	27.41 ± 2.72 ^abAB^	25.73 ± 1.56 ^abcB^	28.26 ± 2.44 ^aA^			
B (r)	−0.48 (−0.54)	−0.16 (−0.23)	−0.09 (−0.10)	−0.27 (−0.25)	−0.49 (−0.47)	0.13 (0.12)			
C *							0.002 *	0.000 *	0.005 *
Day 1	36.90 ± 4.20 ^abA^	35.23 ± 2.15 ^bA^	37.97 ± 2.71 ^abA^	39.94 ± 1.34 ^aA^	39.23 ± 3.19 ^abA^	37.76 ± 3.45 ^abA^			
Day 4	34.34 ± 1.38 ^aAB^	35.27 ± 1.46 ^aA^	34.66 ± 0.98 ^aB^	33.93 ± 2.18 ^aB^	33.89 ± 1.02 ^aB^	34.15 ± 1.23 ^aB^			
Day 7	33.37 ± 2.22 ^bB^	33.22 ± 4.36 ^bA^	36.33 ± 3.64 ^abAB^	37.42 ± 3.12 ^abA^	35.07 ± 2.03 ^abB^	38.12 ± 3.13 ^aA^			
B (r)	−0.59 (−0.47)	−0.34 (−0.28)	−0.27 (−0.23)	−0.42 (−0.31)	−0.69 (−0.54)	0.06 (0.04)			
Colour (Skin)									
L *							0.002 *	0.000 *	0.000 *
Day 1	54.87 ± 3.88 ^aB^	57.56 ± 3.28 ^aA^	54.53 ± 2.27 ^aA^	56.80 ± 2.03 ^aA^	49.51 ± 4.41 ^bB^	54.36 ± 3.11 ^aA^			
Day 4	58.68 ± 1.10 ^aA^	51.88 ± 5.41 ^cAB^	54.86 ± 0.84 ^abcA^	55.48 ± 5.33 ^abcA^	57.50 ± 0.80 ^abA^	52.99 ± 3.55 ^bcA^			
Day 7	52.88 ± 2.27 ^aB^	50.54 ± 6.55 ^abB^	45.55 ± 3.92 ^bB^	45.32 ± 7.10 ^bB^	46.01 ± 5.00 ^abB^	50.29 ± 4.04 ^abA^			
B (r)	−0.33 (−0.23)	−1.17 (−0.49)	−1.50 (−0.73)	−1.91 (−0.66)	−0.58 (−0.24)	−0.68 (−0.42)			
a *							0.006 *	0.000 *	0.000 *
Day 1	1.27 ± 0.29 ^bB^	1.06 ± 0.34 ^bB^	1.20 ± 0.27 ^bB^	1.10 ± 0.16 ^bB^	2.81 ± 1.65 ^aA^	1.23 ± 0.41 ^bA^			
Day 4	1.54 ± 0.44 ^aAB^	1.85 ± 0.46 ^aA^	1.70 ± 0.51 ^aB^	1.69 ± 0.33 ^aB^	1.59 ± 0.27 ^aA^	1.88 ± 0.54 ^aA^			
Day 7	1.95 ± 0.57 ^abA^	1.47 ± 0.37 ^bA^	2.91 ± 0.91 ^abA^	3.14 ± 1.17 ^aA^	2.39 ± 0.83 ^abA^	2.20 ± 1.88 ^abA^			
B (r)	0.11 (0.55)	0.08 (0.42)	0.29 (0.75)	0.34 (0.77)	−0.07 (−0.15)	0.16 (0.34)			
b *							0.090	0.000 *	0.000 *
Day 1	2.62 ± 0.38 ^bcB^	2.46 ± 0.87 ^bcB^	1.58 ± 0.48 ^cC^	2.67 ± 0.54 ^bcB^	4.99 ± 2.13 ^aA^	3.17 ± 0.41 ^bA^			
Day 4	3.43 ± 0.75 ^aA^	3.34 ± 0.84 ^aAB^	3.42 ± 0.89 ^aB^	3.12 ± 1.12 ^aB^	3.07 ± 0.73 ^aB^	4.06 ± 1.08 ^aA^			
Day 7	3.78 ± 0.46 ^aA^	3.68 ± 0.89 ^aA^	5.05 ± 1.03 ^aA^	5.38 ± 1.34 ^aA^	3.89 ± 1.25 ^aAB^	4.11 ± 3.25 ^aA^			
B (r)	0.19 (0.66)	0.20 (0.57)	0.58 (0.87)	0.45 (0.71)	−0.18 (−0.28)	0.16 (0.20)			
C *							0.045 *	0.000 *	0.000 *
Day 1	2.93 ± 0.30 ^bB^	2.69 ± 0.92 ^bB^	2.02 ± 0.40 ^bC^	2.89 ± 0.53 ^bB^	5.74 ± 2.66 ^aA^	3.41 ± 0.49 ^bA^			
Day 4	3.76 ± 0.85 ^aA^	3.83 ± 0.92 ^aA^	3.83 ± 0.99 ^aB^	3.60 ± 0.98 ^aB^	3.46 ± 0.75 ^aB^	4.48 ± 1.19 ^aA^			
Day 7	4.27 ± 0.62 ^aA^	4.02 ± 0.45 ^aA^	5.84 ± 1.33 ^aA^	6.32 ± 1.36 ^aA^	4.57 ± 1.48 ^aAB^	4.69 ± 3.71 ^aA^			
B (r)	0.19 (0.66)	0.20 (0.57)	0.58 (0.87)	0.45 (0.71)	−0.18 (−0.28)	0.16 (0.20)			

Values are expressed as means ± SD. Lowercase letters indicate significant differences between treatments within each day (*p* < 0.05), whereas significant differences between days for individual treatments are indicated with uppercase letters (*p* < 0.05). The effect of two factors, treatments (T) and storage time (ST), and their interaction (T × ST) were determined by two-way ANOVA with Tukey multiple post hoc tests at *p* < 0.05. Significant differences were labelled with (*). ^1^ The relationship between storage day and parameter in individual treatments is expressed in Pearson’s correlation coefficient (r) and the unstandardised regression coefficient (B).

## Data Availability

The data presented in this study are available in the article. The data presented in this study are available on request from the corresponding author.

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
