# Peer review of "Efficacy Evaluation of Chlorine Dioxide and Hypochlorous Acid as Sanitisers on Quality and Shelf Life of Atlantic Salmon (Salmo salar) Fillets"

_foods, 2024, doi:10.3390/foods13193156_

Round 1
Reviewer 1 Report
Comments and Suggestions for Authors
Several questions arise from the results that could be addressed to provide a more comprehensive explanation in the results and discussion section:
-
Lines 285-289: It would be beneficial to explain QIM within the methodology section, clearly detailing how it was interpreted.
-
On what basis did the authors choose concentrations of 5-10 ppm for ClOâ‚‚ and 50 and 100 ppm for HOCl? Additionally, why was only 5 ppm of ClOâ‚‚ used with ice treatment? This should be explicitly explained in the methodology.
-
Why did both ClOâ‚‚ and HOCl treatments initially show significant reductions in drip loss during storage but failed to maintain this effect consistently over the entire 7-day period?
-
Given that MDA levels exceeded acceptable limits throughout the storage period, what adjustments could be made to the application of ClOâ‚‚ and HOCl to better control lipid oxidation?
-
How do the observed microstructural changes, such as increased interfibrillar spaces, correlate with the overall texture preservation in both sanitizer treatments?
-
Why did the ClOâ‚‚ and HOCl treatments fail to prevent microbial counts from exceeding permissible levels by day 7, despite initial reductions in bacterial load?
-
According to Figure 6, the control solution appears to be the most effective in slowing down microbial growth. A thorough discussion should be included regarding the effectiveness of these sanitizers as an antimicrobial agent.
Author Response
Dear Reviewer 1:
My coauthors and I greatly appreciate the encouraging, critical, and constructive feedback provided by the reviewer. The comments were thorough and invaluable in improving the manuscript. We strongly believe that these suggestions have significantly enhanced the scientific value of the revised version. All feedback has been carefully considered and fully incorporated into the manuscript. We are submitting the corrected version with the reviewer’s suggestions applied. Below are our responses to the reviewer’s comments and how they have been addressed in the revised manuscript.
Comment 1: Lines 285-289: It would be beneficial to explain QIM within the methodology section, clearly detailing how it was interpreted.
Response 1: Thank you for your feedback, more in depth details have been added to the QIM methodology section. We have as well reduce lengthiness of details regarding QIM in result and discussion.
Comment 2: On what basis did the authors choose concentrations of 5-10 ppm for ClOâ‚‚ and 50 and 100 ppm for HOCl? Additionally, why was only 5 ppm of ClOâ‚‚ used with ice treatment? This should be explicitly explained in the methodology.
Response 2: The dosages of treatments were chosen based on manufacturer recommendation to be utilised in seafood products. We have added this information to the methodology section. Regarding the usage of ClOâ‚‚ ice, this idea was a preliminary investigation on whether there is a better sanitizing system in seafood, using current practice in poultry sector as reference (ice infusion in retail display).
As there is currently a lack of information available, only one dosage was chosen to compare the impact of introductory method. Moreover, since the primary objective is to compare the two sanitisers rather than comparing introductory system, we believe the comparison of different introductory method with same dosage (5 ppm) would be sufficient to answer this secondary research question while adding diversity and depths to current study.
Comment 3: Why did both ClOâ‚‚ and HOCl treatments initially show significant reductions in drip loss during storage but failed to maintain this effect consistently over the entire 7-day period?
Response 3: Thank you for your question, we have hypothesized in the manuscript that reduce drip loss was due to reducing physical constraints which improve fluid retention. As shelf life continue to progress, increase protein degradation and gapping occurred, as evidenced by TVB-N and histological data. This might have led to the loss/weakening of muscle structure- or protein- dependent effects, thus, result in a failure of significant drip loss reduction.
Comment 4: Given that MDA levels exceeded acceptable limits throughout the storage period, what adjustments could be made to the application of ClOâ‚‚ and HOCl to better control lipid oxidation?
Response 4: In our study we have identified that CIO2 ice at the same dosage shown to have no significant increase in MDA compared to dip treatment. This means that introductory method could have a great impact on MDA. Thus, we suggest that exploration of infused-ice treatment at different dosage or testing of HOCl usage on ice to be the next step in identifying best practice of using tested sanitizer. Additionally, since introductory method shown to have great impact on MDA, researching on optimal introductory method especially borrowing ideas from other animal sector could improve control in lipid oxidation.
Not only this, perhaps using current sanitizer in hurdle application, such as the use of preservative with antioxidant effects (e.g. chitosan) could preserve shelf life while having better control on lipid oxidation. These are some examples of adjustments that we hypothesized based on result from current study.
Comment 5: How do the observed microstructural changes, such as increased interfibrillar spaces, correlate with the overall texture preservation in both sanitizer treatments?
Response 5: Histological changes on day 1 greatly reflect textural changes, such as increase in chewiness and gumminess, as evidenced by increase interfibrillar spaces in all CIO2 treatments.
Treatment degradation evidenced in histological result, especially in day 7, is as well possibly related to retardation in increase of gumminess, chewiness and hardness, possibly due to increase interfibrillar spaces, which in terms allow moisture to be withhold, increase plasticity and reducing hardness.
The results are in alignment of both analyses; however, correlation is uncertain due to quantitative and qualitative nature of two analyses.
Comment 6: Why did the ClOâ‚‚ and HOCl treatments fail to prevent microbial counts from exceeding permissible levels by day 7, despite initial reductions in bacterial load?
Response 6: From figure 6, increase in microbial load from treatments were observed since day 1, which indicate that not only does treatment failed to reduce microbial load more effectively than saline, the oxidative damage done to salmon fillet instead increase bacteria load. Possibly due to increase in nutrient available from cell lyses, this is not unexpected, especially since histology result shown an increase in interfibrillar space.
Comment 7: According to Figure 6, the control solution appears to be the most effective in slowing down microbial growth. A thorough discussion should be included regarding the effectiveness of these sanitizers as an antimicrobial agent.
Response 7: Control was sanitized with brine solution, which could have a long-lasting anti-microbial effect. Our result does not indicate that the sanitizers are not functioning, but rather less effective compared to saline.
The effectiveness of sanitizer was probably lower due to combination of oxidative cell damage which increase available nutrients for microbes and volatile nature of chlorine, which failed to provide a lasting effect that sanitize the food commodity.
However, this as well proven that perhaps dipping was not the most ideal introductory system, which align with above, such as introducing sanitizer with ice infusion to contour volatile nature of chlorine. In which it shown to have comparable result to control and is worth to be investigated next step.

Reviewer 2 Report
Comments and Suggestions for Authors
This manuscript investigated the physical, chemical, and microbiological changes in simulated retail Atlantic salmon fillets subjected to two disinfectants (ClO2 and HOCl). The study content is substantial, but the analysis of results requires improvement. Here are some comments:
1. Clarify the storage times of salmon fillets in Figure 1B. In lines 242-243, the statement needs revision as per Figure 1B; except for the HOCl (100 ppm) treatment, drip loss does not show significant variation in all other treatments. Similarly, review the description for Figure 4B.
2. Verify if there is a significant difference in drip loss for salmon fillets stored for 4 d and 7 d of control treatment in Figure 1A. Similarly, check the significance results marking in Figures 4A and 4C.
3. Improve the annotation of Figure 2; there are no ABCDEF markings in Figure 2.
4. In line 434, revise TBARS to MDA.
5. It is recommended to add a correlation analysis between the indicators.
6. In line 593, the sentence is not clear. Consider deleting line 607.
7. Ensure the reference format is consistent, and the species names in the references should be italicized.
8. What practical implications do the study results have for the preservation of salmon fillets?
9. Provide a manuscript with no track changes for the second review.
Author Response
Dear Reviewer 1:
My coauthors and I greatly appreciate the encouraging, critical, and constructive feedback provided by the reviewer. The comments were thorough and invaluable in improving the manuscript. We strongly believe that these suggestions have significantly enhanced the scientific value of the revised version. All feedback has been carefully considered and fully incorporated into the manuscript. We are submitting the corrected version with the reviewer’s suggestions applied. Below are our responses to the reviewer’s comments and how they have been addressed in the revised manuscript.
Comment 1: Clarify the storage times of salmon fillets in Figure 1B. In lines 242-243, the statement needs revision as per Figure 1B; except for the HOCl (100 ppm) treatment, drip loss does not show significant variation in all other treatments. Similarly, review the description for Figure 4B.
Response 1: Original figure 1A represents the effect of the main factors. We have now added individual lettering to show significant differences between treatments within each sampling day (lowercase letters), as well, significant difference between storage day for each treatment (uppercase letters). Regardless of storage time, we have created figure 1B using pooled data to allow simple visualization of comparison between treatments which allow identification of ideal treatment. Additionally, We have now revised figure captions to improve clarity, reworded lines 242-243 with more clarity and added information to line 421-422, especially the rationale behind figure 1B and 4B – to combine the values obtained across storage day to understand the mass effect of sanitizers.
Comment 2: Verify if there is a significant difference in drip loss for salmon fillets stored for 4 d and 7 d of control treatment in Figure 1A. Similarly, check the significance results marking in Figures 4A and 4C.
Response 2: Now we have revised all figures to demonstrate significant difference between treatments within each day, as well between days of the same treatment. Uppercase letter in figure 1A indicate that there is no significant drip loss in control between day 4 and 7. Thank you for your feedback.
Comment 3: Improve the annotation of Figure 2; there are no ABCDEF markings in Figure 2.
Response 3: Thanks for pointing out and now we have revised Figure 2 and added alphabetical letters to indicate significant differences.
Comment 4: In line 434, revise TBARS to MDA.
Response 4: We have now reworded all TBARS to MDA for more consistency and clarity.
Comment 5:It is recommended to add a correlation analysis between the indicators.
Response 5: We have now added Figure 7 which included correlation analyses of all investigated parameters. As well, inserted result into appropriate discussion.
Comment 6: In line 593, the sentence is not clear. Consider deleting line 607.
Response 6: Line 593 has now been reworded and deleted line 607.
Comment 7: Ensure the reference format is consistent, and the species names in the references should be italicized.
Response 7: We have now disconnected the current manuscript with EndNote and corrected them individually.
Comment 8: What practical implications do the study results have for the preservation of salmon fillets?
Response 8: We have summarized major and important findings in the graphical abstract. Our manuscript identified that all sanitizers could reduce drip loss, which is related to economic loss during retail. In this article, we have also found that sanitizer protects against deterioration of quality related parameters including texture, color and TVB-N alteration during storage. However, unexpectedly treatments shown to have negative impacts on lipid oxidation and bacteria load, which we suspected that same treatment introduced via different format (e.g. ice infusion) at a higher dosage could be more ideal treatment for future studies to extend shelf life of salmon fillets. All in all, we belief that chlorine dioxide and hypochlorous acid could be a promising solution to replace current sanitizer, however, more research is needed to be done for commercial applications of this preservative. Which could have the practical implications of (1) reduce food loss during storage (2) replace current sanitizer that have significant greater health implications
Comment 9: Provide a manuscript with no track changes for the second review.
Response 9: We have now removed track changes. We apologize for the inconvenience.

Round 2
Reviewer 2 Report
Comments and Suggestions for Authors
The author has addressed all my comments and meticulously revised the manuscript.